# Identification of Novel Pathways Regulated by APE1/Ref-1 in Human Retinal Endothelial Cells

**DOI:** 10.3390/ijms24021101

**Published:** 2023-01-06

**Authors:** Mahmut Mijit, Sheng Liu, Kamakshi Sishtla, Gabriella D. Hartman, Jun Wan, Timothy W. Corson, Mark R. Kelley

**Affiliations:** 1Herman B Wells Center for Pediatric Research, Department of Pediatrics, Indiana University School of Medicine, Indianapolis, IN 46202, USA; 2Department of Medical and Molecular Genetics, Indiana University School of Medicine, Indianapolis, IN 46202, USA; 3Department of Ophthalmology, Eugene and Marilyn Glick Eye Institute, Indiana University School of Medicine, Indianapolis, IN 46202, USA; 4Stark Neurosciences Research Institute, Indiana University School of Medicine, Indianapolis, IN 46202, USA; 5Melvin and Bren Simon Comprehensive Cancer Center, Indiana University School of Medicine, Indianapolis, IN 46202, USA; 6Department of Pharmacology and Toxicology, Indiana University School of Medicine, Indianapolis, IN 46202, USA; 7Department of Biochemistry and Molecular Biology, Indiana University School of Medicine, Indianapolis, IN 46202, USA

**Keywords:** APE1/Ref-1, redox signaling, angiogenesis, inflammation, gene expression, ocular diseases, DNA repair

## Abstract

APE1/Ref-1 (apurinic/apyrimidinic endonuclease 1, APE1 or APEX1; redox factor-1, Ref-1) is a dual-functional enzyme with crucial roles in DNA repair, reduction/oxidation (redox) signaling, and RNA processing and metabolism. The redox function of Ref-1 regulates several transcription factors, such as NF-κB, STAT3, HIF-1α, and others, which have been implicated in multiple human diseases, including ocular angiogenesis, inflammation, and multiple cancers. To better understand how APE1 influences these disease processes, we investigated the effects of *APEX1* knockdown (KD) on gene expression in human retinal endothelial cells. This abolishes both DNA repair and redox signaling functions, as well as RNA interactions. Using RNA-seq analysis, we identified the crucial signaling pathways affected following *APEX1* KD, with subsequent validation by qRT-PCR. Gene expression data revealed that multiple genes involved in DNA base excision repair, other DNA repair pathways, purine or pyrimidine metabolism signaling, and histidine/one carbon metabolism pathways were downregulated by *APEX1* KD. This is in contrast with the alteration of pathways by *APEX1* KD in human cancer lines, such as pancreatic ductal adenocarcinoma, lung, HeLa, and malignant peripheral nerve sheath tumors. These results highlight the unique role of APE1/Ref-1 and the clinical therapeutic potential of targeting APE1 and pathways regulated by APE1 in the eye. These findings provide novel avenues for ocular neovascularization treatment.

## 1. Introduction

Aberrant neovascularization (NV) is a hallmark of major, blinding ocular diseases including neovascular “wet” age-related macular degeneration (nAMD), proliferative diabetic retinopathy (PDR), and retinopathy of prematurity (ROP) [1]. Collectively, these diseases are major causes of visual impairment [2]. Recent years have seen the elucidation of pathways involved in angiogenesis and the advent of therapies targeting vascular endothelial growth factor (VEGF) signaling. However, despite these successes, a significant fraction of patients are refractive to anti-VEGF therapy [3]. Thus, there is an urgent, unmet need to identify novel antiangiogenic targets leading to more effective therapies.

A key feature of nAMD is choroidal neovascularization (CNV), the growth of blood vessels into the subretinal space in the macular area, while ROP and PDR are associated with NV from the retinal vasculature [1,4]. CNV damages central vision, causing blurred vision and blind spots. Blockade of VEGF is the standard of care for nAMD and widely used for PDR and ROP using biologics such as bevacizumab, ranibizumab, aflibercept, or the newer agents brolucizumab and faricimab [5]. However, solely targeting VEGF is often not sufficient to prevent NV. Other treatment hurdles include the intravitreal delivery system, which carries the small but important risk of endophthalmitis, and patients report that they would prefer to avoid injections if oral or eyedrop therapies were available [6]. Research into additional targets of angiogenesis in the eye to reduce NV-induced vision loss and blindness is needed.

Beyond angiogenesis, inflammation also plays a role in the development and pathogenesis of many ocular diseases, such as AMD, ROP, and PDR. In the vitreous of patients affected with retinopathies, increased levels of pro-inflammatory molecules, including TNF-α, IL-6, IL-8, and others, have been observed [7,8].

APE1/Ref-1 is a dual-functional enzyme that has been implicated in multiple human diseases [9,10,11]. The apurinic/apyrimidinic endonuclease activity of APE1 is critical for repairing damaged DNA and/or RNA to maintain genome stability [12,13]. The redox activity of Ref-1 regulates numerous transcription factors (TFs), including NF-κB, STAT3, HIF-1α, and others, all known to be involved in ocular angiogenesis and inflammation [9,10]. APE1 redox signaling activity reduces cysteine residues in its target proteins through a protein–protein interaction and a unique mechanism of action distinct from other redox proteins, such as thioredoxin. The redox signaling function is distinct from its DNA repair activity. The critical cysteine (Cys) in APE1, Cys65, is involved in the electron transfer reduction of the target TF Cys(s) and can be mutated to block the redox function of APE1 without affecting the DNA repair capabilities. Likewise, mutations of amino acids required for DNA repair activity have no effect on the redox action. Reduction of TFs by APE1 stimulates them to bind to their DNA targets, thereby participating in a wide spectrum of cellular functions, including inflammation and neovascularization. Additional functions of APE1 include degradation of RNA and cleavage of RNA-containing abasic sites [9,14]. Moreover, APE1 also regulates multiple oncogenic microRNAs (miRNA) directly or indirectly through regulating miRNA-TFs networking, which affects gene expression profiles [13,15,16]. Taken together, APE1 has crucial functions in the response to DNA damage and as a master regulator of disease-relevant transcription.

In vitro work has shown that under normal circumstances, endothelial cells depend on APE1 redox activity for proliferation, migration, and angiogenesis. Previous studies have demonstrated that disruption of APE1 redox signaling by the small molecule inhibitor APX3330 reduced proliferation, migration, and angiogenesis in choroidal endothelial cells [17]. APX3330 is currently in a Phase 2b trial (NCT04692688) as an oral drug for diabetic retinopathy (DR) and diabetic macular edema (DME). In cases of cellular stress in the eye, APE1 is expressed in response to tissue inflammation [18]. Furthermore, APE1 is upregulated during murine retinal development, and highly expressed in retinal pigment epithelium cells, retinal pericytes, choroidal endothelial cells, and retinal endothelial cells [18,19]. Recently, APE1 has been identified as overexpressed in the context of retinal and choroidal neovascularization as well [20,21].

In this study, we sought to further explore and discover the roles of APE1 in human retinal endothelial cells (HRECs) through siRNA knockdown (KD) of the *APEX1* gene and subsequent assessment of gene expression via RNA-seq. KD of *APEX1*, which removes both the DNA repair and redox signaling functions, as well as RNA interactions, led to differentially expressed genes (DEGs) clustered around DNA base excision repair (BER), other DNA repair pathways, and multiple metabolism pathways in HRECs. This is in contrast with the gene regulation patterns frequently seen in cancer cells with *APEX1* KD. These results highlight the unique functions and therapeutic potential of APE1 in the eye.

## 2. Results

### 2.1. Differential Gene Expression Analysis for APEX1 KD in HRECs

Differential analysis was conducted on genome-wide gene expression between *APEX1* KD in HRECs and two control groups, scrambled siRNA (SCR) and/or mock transfection with Lipofectamine only (MOCK), resulting in 3621 and 4247 DEGs, respectively, given the cutoffs used (see Section 4), as illustrated in Figure 1A,B. The fold changes (FCs) of gene expression upon *APEX1* KD were quite consistent compared with SCR and MOCK, with a high correlation coefficient of 0.91 (Figure 1C). Moreover, significant overlaps (*p* < 2.2 × 10^−16^) were observed between different control comparisons for both upregulated DEGs (Figure 1D) and downregulated ones (Figure 1E), indicating that the majority of DEGs were recognized based on the comparison to both SCR and MOCK. We defined these DEGs confirmed by both controls as the genes responding to *APEX1* KD in HRECs.

To determine the functional roles of DEGs, we performed functional enrichment analysis on KEGG pathways using DAVID [22,23]. The results illustrated that upregulated DEGs were enriched in pathways related to systemic lupus erythematosus, p53 signaling, FoxO signaling, and ribosomes (Figure 1F); pathways associated with DNA replication, cell cycle, mismatch repair, glycolysis/gluconeogenesis, and metabolism were significantly overrepresented in suppressed DEGs (Figure 1G).

### 2.2. Multiple Crucial Signaling Pathways Are Dysregulated Following APEX1 KD in HRECs

We previously demonstrated that 2837 genes’ expressions are significantly changed following *APEX1* KD in a human PDAC line, which resulted in dysregulation of multiple pathways, including the EIF2 signaling and mechanistic target of rapamycin pathways, plus several mitochondrial-related pathways [24]. Here, in the case of HRECs, we identified that DEGs upon *APEX1* KD were most often seen in the pathways of base excision repair (Figure 2A), one carbon (1C) metabolism mediated by folate co-factor (Figure 2B), purine metabolism (Figure 2C), and pyrimidine metabolism (Figure 2D).

### 2.3. Expression Levels of Selected Genes from RNA-seq upon APEX1 KD

We observed that all genes selected from the RNA-seq analysis for validation were expressed in HRECs. However, the expression levels of these genes were varied upon basal condition. The mRNA abundances were evaluated based on the threshold cycle (C_t_) value in the real-time PCR (qRT-PCR) using the same amount of HREC cDNA as input. The C_t_ values of each gene were normalized to the reference gene *BACT* (encoding β-actin). The final readout of the C_t_ value indicates the abundance of expression levels of the gene of interest upon basal condition (Appendix A).

We then performed a conditional *APEX1* knockdown (KD) experiment (KD efficiency >80%) in HRECs to validate the differentially expressed genes from RNA-seq analysis (Figure 3A). Multiple genes associated with several key signaling pathways were significantly affected by *APEX1* KD (Figure 3B–F). In particular, the gene panels in base excision repair and DNA repair pathways (*NEIL3*, *POLB*, *POLD1*, *POLD2*, *POIE2*, *PARP1*, *XRCC1*, *FEN1*, *RFC5*), purine or pyrimidine metabolism signaling (*ADCY7*, *DPYD*, *TK1*, *ATIC*, *DHFR*, *POLA1*), histidine/one carbon metabolism pathways (*ALDH3A1*, *MTFMT*), and glycolysis (*LDHA*, *GALM*, *MNPP1*) were significantly downregulated following *APEX1* KD (SCR vs. *APEX1* KD, *p* < 0.05) (Figure 3B–E, respectively). We also demonstrated that *HMOX1* was significantly upregulated, while *BCL2A1* and *CCNE2* went down (SCR vs. *APEX1* KD, *p* < 0.05) (Figure 3F). Again, our data support that knocking down *APEX1* resulted in the perturbation of multiple crucial signaling and DNA repair pathways in HRECs.

### 2.4. APEX1 Affects Diverse Sets of Genes in HRECs and Different Cancer Cell Lines

To understand whether the effects of *APEX1* KD in retinal endothelial cells are consistent across cell types, we selected four types of cancer cells (PDAC, lung, HeLa, and malignant peripheral nerve sheath tumor (MPNST)) for comparison based on available published datasets. Figure 4A lists 58 downregulated DEGs involved in four KEGG pathways for HRECs (from Figure 1 and Figure 2). We observed diverse gene expression alterations across different cell types. We further defined DEGs in these cancer cells (see Section 4) and compared the overlap between them (Figure 4B–E) and the DEGs identified in HRECs. The overlaps of DEGs were quite limited, indicating that APE1 might play various roles in different cell types via cell-specific target genes. Regardless of statistical significances in other cell lines, we focused on DEGs recognized in HRECs only. The DEGs can be classified into four groups given the direction of their gene expression changes in all five cell types, including all up, up mixed, down mixed, and all down (Figure 4F). In line with the observation on the overlaps of DEGs shown in Figure 4B–E, the majority of DEGs in HRECs showed opposite or mixed FC directions in two groups, up mixed and down mixed. However, a smaller portion of DEGs were found with consistent alterations of gene expression in HRECs and the four cancer cell lines (Figure 4F), including 96 DEGs in all up and 301 DEGs in all down, suggesting potentially common functions of APE1 in all these cell types. Indeed, many genes involved in apoptosis signaling pathways were upregulated in all five cell types in the absence of *APEX1* (Figure 4G), such as *HMOX1*, *TNFRSF100*, *PLAUR*, *ZMAT3*, *CASP2*, *IL1A*, *NACC2*, *ACVR1*, *DAP*, and *EIF2AK3.* In addition, DAVID functional enrichment analysis explored gene ontology (GO) terms and KEGG pathways (Figure 4H) that were significantly overrepresented in DEGs in the group of all down for all five cell types. For example, genes associated with cell cycle, DNA repair, nucleic acid binding, DNA conformational change, and some metabolic pathways were downregulated after *APEX1* KD. This implies that APE1 may orchestrate these biological functions and pathways by regulating common targets independent of the cell types.

## 3. Discussion

In this study, we investigated the effects of *APEX1* KD on gene expression in HRECs to identify pathways regulated by *APEX1* that might contribute to the therapeutic effect of blocking APE1 function. We also contrasted these findings with *APEX1* KD in various human cancer cell types to highlight the unique and shared therapeutic potential of APE1 in ocular diseases. RNA-seq and pathway enrichment analysis revealed that repressed pathways in response to *APEX1* KD in HRECs included genes involved in DNA replication, cell cycle, glycolysis, and several metabolic pathways. Upregulated pathways in HRECs included genes involved in systemic lupus erythematosus, p53 signaling, FoxO signaling, and ribosomes. Further analysis using qRT-PCR validated DEGs that are involved in DNA repair, glycolysis, purine or pyrimidine metabolism, and 1C metabolism.

Because APE1 is a well-characterized AP (apurinic/apyrimidinic) endonuclease involved in DNA BER, it is not surprising that some of the top downregulated genes in response to *APEX1* KD are involved in BER, including *XRCC1*, *FEN1*, *POLB*, and *PARP1.* BER is the primary DNA repair pathway that acts on AP sites that are created following the removal of alkylated or oxidated bases by monofunctional or bifunctional DNA glycosylases, as well as RNA interactions such as degradation in concert with NPM1 [25]. Though APE1 has been shown to interact with NPM1 in previous works [26], our results demonstrate that KD of *APEX1* does not significantly alter expression of NPM1 in HRECs, PDAC, lung, and MPNST cells, suggesting that RNA degradation/processing is not impacted by the KD of *APEX1* in these cell lines under the experimental design used. In BER, a monofunctional (e.g., uracil DNA glycosylase (UNG) or methylpurine glycosylase (MPG)) or bi-functional glycosylase (e.g., OGG1, NEIL1) first excises the damaged base [27]. APE1 initiates repair of the AP sites by hydrolyzing the phosphodiester backbone 5′ to the AP site, allowing for DNA polymerase β to remove the 5′dRP (deoxyribonucleotide-phosphate) and incorporate correct nucleotides [14,28]. Subsequently, DNA ligase III and XRCC1 complete the repair [29]. XRCC1 acts as a scaffold to modulate various steps in the BER pathway by providing a link between the incision and sealing steps in BER [28]. XRCC1 directly binds with FEN1, DNA polymerase β, and APE1 to facilitate its AP endonuclease activity [30,31,32,33,34,35]. The association of XRCC1 and APE1 could serve to facilitate the processing of abasic sites by APE1 [28]. In short-patch BER, APE1, polymerase β, and XRCC1 complex with poly-ADP-ribose-polymerase (PARP) to repair lesions [25]. PARP and FEN1 are involved in long-patch BER and replacement strand DNA synthesis, and APE1 directly binds with FEN1 to enhance FEN1 activity to coordinate long-patch BER and prevent premature cleavage [25,33,35]. Together, our gene expression findings indicate that APE1 coordinates and is involved in most of the steps of BER; alterations in its cellular level impact other important genes in the BER pathway [36]. This contrasts with what has been observed in cancer studies [24,37].

In addition to DNA repair, various metabolic pathways were dysregulated in response to *APEX1* KD in HRECs. Specifically, genes involved in glycolysis were strongly downregulated following siRNA KD of *APEX1* in HRECs, implying that APE1 may participate in regulating cellular glucose homeostasis and metabolism in the retina. A recent publication demonstrated that *APEX1* KD in PDAC cells downregulated genes involved in metabolic pathways, including glycolysis, suggesting a universal function of APE1 in regulating glycolysis [38]. Glucose homeostasis is regulated by glycolysis, the TCA cycle, and OXPHOS, and dysregulation of glucose homeostasis contributes to the pathogenesis of blinding NV eye diseases such as diabetic retinopathy (DR) and AMD [39]. In the retina, glycolysis is essential for endothelial cell (EC) proliferation, as the retina metabolizes most glucose through anaerobic glycolysis to promote the rapid production of ATP with faster kinetics and produce lactate and other intermediates necessary for proliferating cells to generate amino acids, lipids, and nucleotides [40,41,42]. ECs rely on glycolysis instead of OXPHOS to produce ATP during angiogenesis, even in the presence of oxygen [41,43,44,45,46]. This metabolic alteration, known as the Warburg effect, is also utilized by cancer cells to allow for their proliferation [39,47]. ScRNA-seq revealed that angiogenic ECs become enriched in glycolysis genes compared to quiescent ECs [41,48]. The energy and lactate produced through glycolysis is used by ECs to switch from a quiescent to angiogenic phenotype, support cell proliferation and migration, and allow for increased filopodial formation that is essential for tip cell differentiation [40,41]. The downregulation of glycolytic genes in response to *APEX1* KD that we observed in HRECs may suggest that APE1 plays an angiogenic role in the retina by promoting glycolytic activity.

*LDHA*, the glucose-usage-related gene, was one of the strongly downregulated glycolytic genes that we identified in response to *APEX1* KD in HRECs; similar results were seen previously in PDAC cells [38]. *LDHA* is the last enzyme in the glycolytic pathway and catalyzes the conversion of pyruvate to lactate. Inhibition of *LDHA* gene expression suppresses tumor growth and impairs vascularization, indicating a functional role of *LDHA* in aberrant NV [49,50,51,52,53]. LDHA is essential for microvascular EC VEGF production during angiogenesis, and inhibition of LDHA reduces the number of tip cells, indicating that LDHA is necessary for tip cell differentiation and maintenance [40,54,55]. Inhibition of LDHA activity or decreasing *LDHA* gene levels suppresses cancer cell growth both in vitro and in vivo [50,51]. This evidence, together with our new findings, suggests that APE1 may play a role in tip cell differentiation and microvascular EC growth through glycolysis and the VEGF signaling pathway. Lactate also serves as a signaling molecule in angiogenesis; HIF-1α is stabilized when lactate levels are high, and HIF-1α stimulates angiogenesis [53]. APE1 allows for transcriptional activation of HIF-1α [56,57]. This, in combination with our new findings, may indicate that APE1 regulates HIF-1α more intricately than previously thought. Together, this evidence suggests that APE1 plays a role in NV and tumor cell growth through the regulation of lactate formation via glycolysis. This is supported by preclinical data demonstrating that blocking just APE1′s redox signaling function in the retina blocks ocular angiogenesis [18,19,58]. These findings have led to a Phase 2b clinical trial in DR and DME with the APE1 redox signaling small molecule inhibitor APX3330 [59].

Proper cell division for proliferation requires an adequate supply of nucleotides for DNA and RNA synthesis. These nucleotides can be produced by leveraging amino acids and other small molecules to build purine and pyrimidine rings [39,47]. Purine metabolism is implicated in several NV eye diseases, and previous work has identified aberrant purine metabolites in the plasma of DR patients, indicating dysfunctional purine metabolism during disease to support EC proliferation [60,61]. Top downregulated genes involved in purine/pyrimidine metabolism following *APEX1* KD in HRECs include *DPYD* (dihydropyrimidine dehydrogenase), the rate-limiting enzyme for catabolism of pyrimidines; *TK1* (thymidine kinase 1), which serves to maintain sufficient dNTPs for DNA replication and supports metabolic reprogramming; and *ATIC* (5-aminoimidazole-4-carboxamide ribonucleotide formyltransferase/IMP cyclohydrolase), which catalyzes the last two steps of de novo purine biosynthetic pathways. ATIC is upregulated in cancer cells, and knockdown decreases cell proliferation and migration [62,63]. TK1 activity is correlated with cell proliferation and cancer pathogenesis [64,65]. Cells also require 1C units for nucleotide synthesis to support high proliferative rates. In addition, 1C contributes to many downstream pathways known to benefit cancer cell survival [66]. Because our results demonstrated downregulation of 1C metabolism genes in response to *APEX1* KD, APE1 may further enhance cell proliferation by regulating the folate co-factor. These data indicate that APE1 may leverage 1C and purine/pyrimidine metabolism to increase cell proliferation and enhance aberrant NV as seen in blinding NV eye diseases. These findings are not only highly relevant to NV; to our knowledge, this is the first time this relationship between APE1 expression and the one-carbon pool by folate and purine/pyrimidine metabolism has been directly observed, and it is distinct from *APEX1* KD findings in cancer studies. These data also suggest that targeting purine/pyrimidine metabolism could be a potential therapeutic avenue, but additional studies will be needed to ascertain which APE1 function is implicated in this finding.

Interestingly, transcription factors regulated by APE1 have an impact on cellular metabolic reprogramming, including HIF-1α, STAT3, and NRF2 [56,67,68,69]. Ref-1 redox signaling works to activate HIF-1α and STAT3 and deactivate NRF2. A few additional genes had a strong response to *APEX1* KD in HRECs, including the upregulation of *HMOX1.* Previous work has shown that inactivation of APE1 resulted in upregulation of NRF2 target genes like *HMOX1* [67], which may be a cellular response to the blockade of APE1 on various metabolic pathways. *HMOX1* may contribute to wound repair, as previous work demonstrates that it responds to injury and is cytoprotective [70]. Thus, *HMOX1* may be indirectly negatively regulated by APE1 and reflects an overall response of the cells to increased oxidative stress and DNA damage in the absence of APE1, with protective effects on ECs.

The number of DEGs in common between HRECs and the cancer cell lines was quite limited, indicating that APE1 may play differential roles in different tissues. This underscores the need to assess APE1 function and expression in different systems and tissues to fully understand the role of APE1 in each individual system and disease and not generalize. Out of the various pathways and cell types analyzed, genes involved in cell cycle regulation, DNA repair, nucleic acid binding, DNA conformational change, and double-strand break repair were downregulated in all cell types. This indicates that APE1 has a universal role in DNA repair and regulation of the cell cycle, regardless of cell type. This is not surprising given APE1′s known function as a major and essential DNA repair enzyme. Additionally, genes involved in various metabolic pathways, such as purine and pyrimidine metabolism, were downregulated in all cell types in response to *APEX1* KD with siRNA. This evidence supports that in pro-angiogenic/pro-growth environments, APE1 is active in a variety of different cell types and regulates cellular metabolic pathways and the cell cycle. Upregulated pathways included p53, FoxO, and apoptosis signaling. FoxO1 regulates gluconeogenesis and glycogenolysis and works to maintain endothelial cells in a quiescent state and restrict vascular overgrowth by reducing glycolysis [41]. Thus, APE1 may work to repress the transcriptional activity of FoxO1 to regulate glycolysis and promote vascular overgrowth. Apoptosis signaling is upregulated in the absence of APE1 in all cell types, including the genes *EIF2AK3*, *TNFRSF100*, and *IL1A*. These results may indicate that APE1 is necessary to maintain the integrity of the cell and prevent apoptosis. This is supported by literature evidence demonstrating that complete knockout of *Apex1* is embryonically lethal [71].

Because APE1 is a multifunctional protein with functions in both DNA BER and redox activity, whether the repair or redox activity caused the downregulation of the observed genes after *APEX1* KD is unknown. Moreover, APE1 is a pivotal regulator in RNA processing and metabolism, in which APE1 regulates miRNAs directly or indirectly. This affects various biological processes [13,15,16], so *APEX1* KD could exert its effects (partially) through miRNA modulation; this remains to be explored further. Currently, we hypothesize that the downregulation of DNA repair genes is mainly due to the AP endonuclease activity of APE1, while the downregulation of metabolic genes may be due to the transcriptional redox regulation function of APE1. Experimental data supporting this hypothesis have been published, albeit in relation to cancer cells [38]. However, we cannot conclude that these functions exist without specifically blocking each function in HRECs, and we recognize this as a limitation of the current study. Although APE1 redox-specific inhibitors are widely used and have demonstrated specificity, recent work suggests that repair genes could be under redox control too [72]. Future studies are warranted with use of redox inhibitors such as APX3330 and APX2009 and repair inhibitors such as APE1 repair inhibitor III (ARi3) to decipher how each APE1 function perturbs each individual pathway. However, the APE1 DNA repair inhibitors currently used have off-target effects, have not progressed translationally, and may not be instructive. APE1 RNA processing/metabolism-specific inhibitors could be explored once more data on specificity are elucidated on these molecules. Nonetheless, the findings of this study are enlightening since few studies have knocked down *APEX1* in primary cells such as HRECs as most studies have been done in transformed cancer cell lines.

Other future directions for research include conducting metabolomics after manipulating APE1 in the HRECs and in vivo in the eye, as well as performing proteomics in the eye and HRECs under KD and using specific function APE1 inhibitors to determine if APE1 has the same effects on the identified pathways at the protein level. Such work can fully elucidate the regulation of APE1 on cellular metabolism and its pro-angiogenic effects as seen in NV eye diseases.

## 4. Materials and Methods

### 4.1. Cell Transfection with siRNA

Human retinal endothelial cells (HRECs) were obtained from Cell Systems (Kirkland, WA, USA) and grown in EGM-2 (Lonza, Walkersville, MD, USA) on Attachment-Factor-coated plates (Cell Systems) in a humidified 37 °C incubator with 5% CO_2_. HRECs were used between passages 4 and 7 and checked regularly for mycoplasma contamination. For selective KD of *APEX1* in HRECs, the following siRNAs were used: scrambled (SCR) (5′-CCAUGA GGUCAGCAUGGUCUG-3′, 5′-GACCAUGCUGACCUCAUGGAA-3′) and siAPEX1 (5′-GUCUGGUACGACUGGAGUACC-3′, 5′UACUCCAGUCGUACCAGACCU-3′). All siRNA transfections were performed as previously described [38]. Briefly, 1.5 × 10^5^ cells were plated per well of a 6-well plate and allowed to attach overnight. The next day, Lipofectamine RNAiMAX reagent (Invitrogen, Carlsbad, CA, USA) was used to transfect in the *APEX1* or SCR siRNA at concentrations of 25 nM following the manufacturer’s instructions. A mock transfection condition was also included.

### 4.2. RNA-seq Library Preparation and Sequencing

RNA-seq was performed at the Indiana University Center for Medical Genomics according to their standard protocols. Briefly, total RNA samples were first evaluated for their quantity and quality using an Agilent Bioanalyzer 2100. Then, 100 ng of total RNA was used for library preparation with the KAPA mRNA Hyperprep Kit (KK8581) (Roche, Basel, Switzerland). The library preparation included poly(A) RNA enrichment, RNA fragmentation, cDNA synthesis, ligation of index adaptors, and amplification. Each resulting uniquely dual-indexed library was quantified and quality-assessed by Qubit and Agilent Bioanalyzer, and multiple libraries were pooled in equal molarity. The pooled libraries were sequenced on an Illumina HiSeq 4000 sequencer using the v1.0 reagent kit to generate 100 bp paired-end reads.

### 4.3. Bioinformatics and Data Analyses

The sequencing reads from the RNA-seq generated in-house (e.g., *APEX1* KD and corresponding controls in HRECs) were mapped to the human genome hg38 using STAR (v2.5) with the following parameter: “--outSAMmapqUnique 60” [73]. Uniquely mapped high-quality reads were assigned to the GENCODE 31 genome using the feature Counts (v2.0.1) with the parameters: “-s 2 –p –Q 10 -O” [74]. The lowly expressed genes were filtered out if they did not pass the cutoff of read counts of more than 10 in at least 4 of the samples. The expression profiles of other genes were normalized using the TMM (trimmed mean of M values) method, then subjected to differential expression analysis based on the comparisons between *APEX1* KD and controls using edgeR (v3.20.8) [75,76]. DEGs were determined with a false discovery rate (FDR) of less than 0.05. Only DEGs identified by both comparisons of *APEX1* KD against the two controls (SCR and MOCK) in HRECs were selected for downstream analysis.

The *APEX1* KD effects in selected cancer cells were evaluated differently depending on the data resources, qualities, and processing methods. (1) The MPNST (malignant peripheral nerve sheath tumor) cell line NF90-8 was received from Dr. Verena Staedtke (Johns Hopkins University), and *APEX1* KD was performed in NF90-8 cells as previously reported [37]. The RNA-seq analysis was conducted in the same manner as for *APEX1* KD in HRECs. The DEGs in MPNST were identified by the cutoff of FDR less than 0.05. (2) The raw data for the gene expression in HeLa cells were retrieved from the E-MEXP-1315 [77]. After the robust multichip average (RMA) normalization for the microarray data, differential expression analysis was performed using the package limma [78]. DEGs were determined with FDR less than 0.01 and absolute value of log_2_FC larger than 1. (3) Gene expression alterations associated with *APEX1* KD for lung cancer cells were downloaded directly from the GEO (GSE74572). DEGs were defined by FDR less than 0.01. (4) The list of DEGs for pancreatic ductal adenocarcinoma (PDAC) data was provided by the authors [79]. The overlaps of DEGs in HRECs and the other four data sets were plotted using UpSetR [80]. The heatmaps for DEGs in the HREC data and selected genes were displayed in the scale of log_2_FC in corresponding data sets using Complex Heatmap [81].

Gene ontology (GO) and KEGG pathway functional analysis were performed by using DAVID on specific gene sets of our interest (e.g., either up- or downregulated DEGs) after *APEX1* KD in HRECs, or DEGs downregulated in all five cell types [22,23]. Significantly overrepresented GO terms or KEGG pathways were selected by the cutoff of FDR < 0.05 and plotted using customized scripts. The pathview software was also used to present specific KEGG pathways and highlight individual genes with corresponding FCs (in log_2_ scale) [82].

### 4.4. Immunoblot

HRECs were lysed in RIPA extraction buffer supplemented with protease and phosphatase inhibitors (Santa Cruz Biotechnology, Santa Cruz, CA, USA) [83]. Denatured samples (20 μg) were subjected to SDS-PAGE. Non-specific binding sites were blocked at room temperature for 1 h with 5% (*w*/*v*) Blotting-Grade Blocker (Bio-Rad Laboratories, Hercules, CA, USA) in Tris-buffered saline (Boston BioProducts, Boston, MA, USA) containing 0.05% (*v*/*v*) Tween-20 (Thermo Fisher, Waltham, MA, USA). Membranes were incubated overnight with the primary antibodies, anti-APEX1 (1:1000) (Novus Biologicals, Centennial, CO, USA; 13B8E5C2), and anti-vinculin (1:1000) (Novus; CP74-100), and then with the peroxidase-conjugated secondary antibody (1:1000) (Bio-Rad, 1706516) for 1 h. Signals were then captured by using a Bio-Rad ChemiDoc imager, and band intensities were analyzed by densitometry using Image Lab software (Bio-Rad).

### 4.5. qRT-PCR

Total RNA was extracted from cells using the Qiagen RNeasy Mini kit (Qiagen, Valencia, CA, USA) according to the manufacturer’s instructions. First-strand cDNA was synthesized from RNA using random hexamers and MultiScribe reverse transcriptase (Applied Biosystems, Foster City, CA, USA). Quantitative PCR was performed using SYBR Green Real-Time PCR master mix (Applied Biosystems) in a CFX96 real-time detection system (Bio-Rad). The relative quantitative mRNA level was determined using the comparative C_t_ method using *BACT* as the reference gene. qRT-PCR cycling conditions were 1 min at 95 °C, 10 min at 95 °C, 15 s at 95 °C, and 1 min at 60 °C for 40 cycles. The primers used for qRT-PCR are detailed in Appendix A. At least three independent experiments were performed. Relative quantity was determined using the 2^−ΔΔCt^ method as previously published [83].

### 4.6. Statistics

All validation experiments were performed independently at least three times. The obtained data were expressed as mean  ±  SEM. Student’s *t*-test or one-way ANOVA were used for calculating statistical significance for qRT-PCR or immunoblot using GraphPad Prism Version 9. The difference was considered statistically significant when *p*  <  0.05.

## Figures and Tables

**Figure 1 ijms-24-01101-f001:**
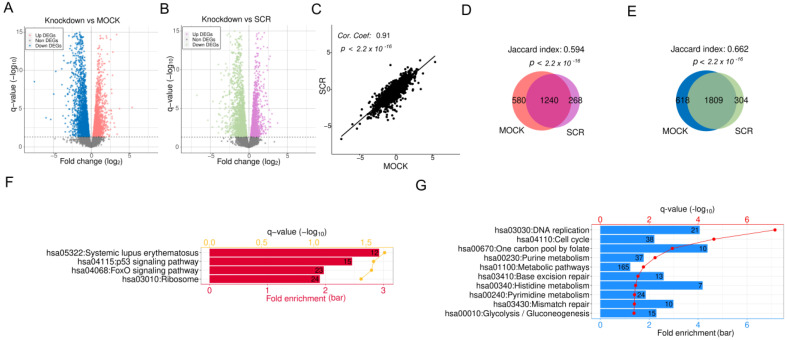
RNA-seq analysis of *APEX1* knockdown (KD) in human retinal endothelial cells (HRECs). (**A**) Volcano plot of *APEX1* KD in HRECs compared with mock transfected (MOCK) HRECs. The pink and blue points represent upregulated and downregulated DEGs, respectively. (**B**) Volcano plot of *APEX1* KD compared with scrambled siRNA-transfected cells (SCR), where the purple and green dots indicate up- and downregulated DEGs, respectively. (**C**) Fold changes (FCs) of gene expression upon *APEX1* KD compared to two controls, MOCK (*x*-axis) and SCR (*y*-axis). (**D**) Overlap of upregulated DEGs between KD vs. MOCK and KD vs. SCR. (**E**) Overlap of downregulated DEGs compared to two controls. Selected KEGG pathways significantly overrepresented in (**F**) upregulated DEGs and (**G**) downregulated DEGs.

**Figure 2 ijms-24-01101-f002:**
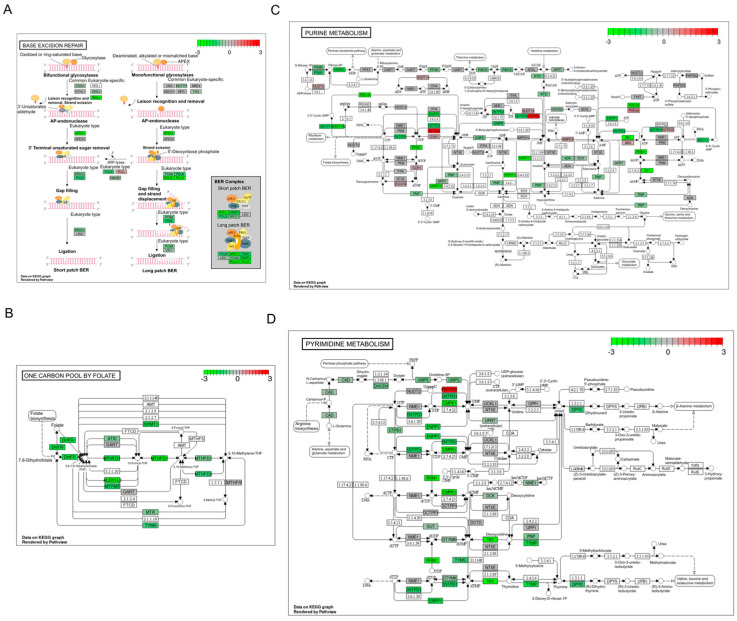
Signaling pathways profoundly affected by *APEX1* KD in HRECs. Color-coded FCs for individual DEGs involved in significantly overrepresented KEGG pathways: (**A**) base excision repair (BER), (**B**) one carbon pool by folate, (**C**) purine metabolism, and (**D**) pyrimidine metabolism.

**Figure 3 ijms-24-01101-f003:**
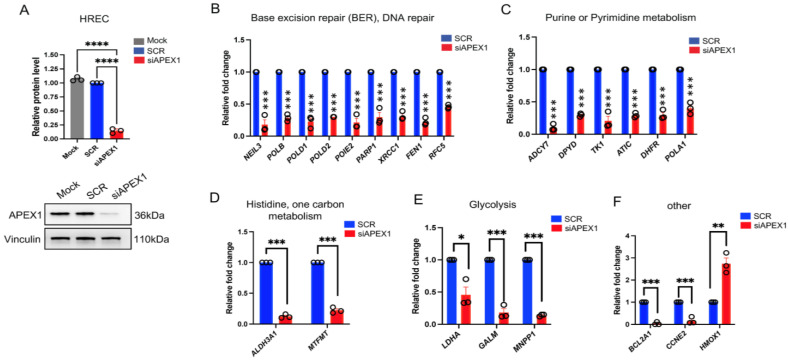
Validation of expression changes of selected genes from RNA-seq upon *APEX1* KD. HRECs were transfected with 25 nM *APEX1* siRNA and scrambled (SCR) control for 48 h. (**A**) The knockdown (KD) efficiency (>80% vs. SCR) was then confirmed by immunoblot. Vinculin was used as loading control. One-way ANOVA, **** *p* < 0.0001. These samples were used for the qPCR target validation. (**B**–**F**) mRNA expression levels of gene panels associated with multiple signaling pathways were significantly downregulated by *APEX1* KD. In particular, the gene panels in base excision repair and DNA repair pathways, purine or pyrimidine metabolism signaling, histidine/one carbon metabolism pathways, glycolysis, and other pathways were assessed by qRT-PCR (Figure 2B–F, respectively). * *p* < 0.05, ** *p* < 0.01, *** *p* < 0.001, Student’s *t*-test. Data expressed as mean ± SEM, n = 3.

**Figure 4 ijms-24-01101-f004:**
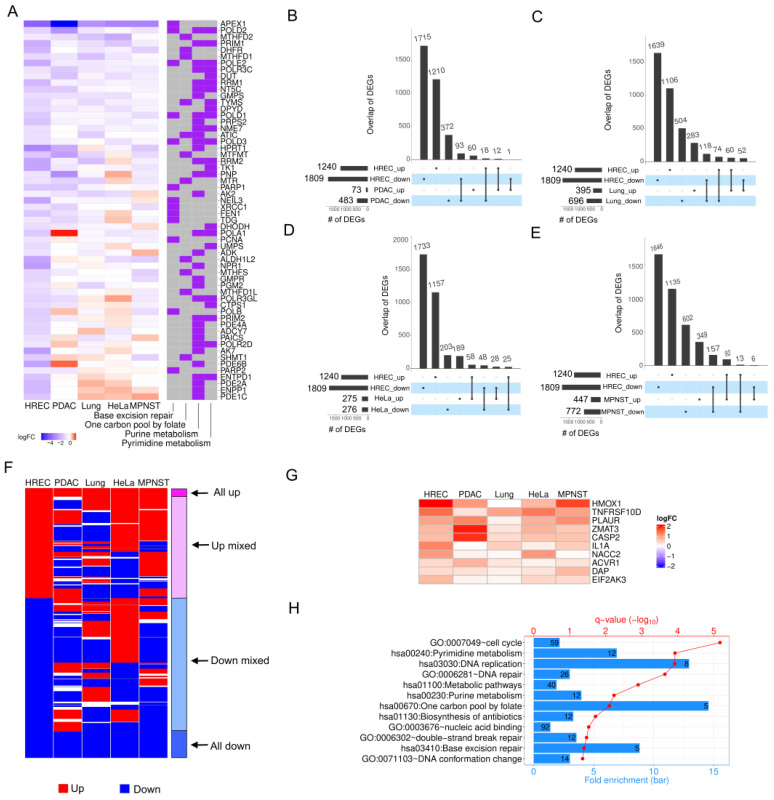
Comparisons of DEGs upon *APEX1* KD in HRECs and four cancer cell datasets, PDAC, lung, HeLa, and MPNST. (**A**) Expression changes in terms of log_2_FC of selected genes in HRECs and four types of cancer cells upon *APEX1* KD. Overlaps of DEGs identified in HRECs and (**B**) PDAC, (**C**) lung cancer cells, (**D**) HeLa cells, (**E**) MPNST. (**F**) Directions of gene expression alterations (red: up, and blue: down) of DEGs upon *APEX1* KD identified in HRECs compared to those in cancer cells. (**G**) FCs of upregulated DEGs in HRECs and cancer cells associated with the apoptosis signaling pathway. (**H**) Gene ontology (GO) and KEGG pathways significantly overrepresented in downregulated DEGs identified in HRECs whose expression levels were decreased in cancer cells as well. “#” is referring to the numbers of differential expressed genes (DEGs).

## Data Availability

The raw data supporting the conclusions of this article can be found in the GEO, accession number is GSE217746.

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
