# Peer review of "Identification of Novel Pathways Regulated by APE1/Ref-1 in Human Retinal Endothelial Cells"

_ijms, 2023, doi:10.3390/ijms24021101_

Round 1

Reviewer 1 Report

This paper is clearly written and the overall experimental plan is sound. I have some major issues that should be addressed:

1.  DEG validation should be confirmed through Western blotting experiments on at least the most interesting genes

2. The author should test whether some of the DEG are regulated through redox based mechanisms by APE1 using the well known APE1 redox inhibitor E3330

3. An emerging field of investigation is the role of APE1 in miRNAs processing. The effect the authors show on DEG could be due to miR-mediated effects. The authors should discuss this aspect.

Author Response

Please find our responses in attached word file.

Reviewer 2 Report

Dear Authors,

interesting results!

An important consideration has to be made: a clearer integration of the clinical implications of these results is important, starting from the abstract and also extensively in the discussion in order to make this paper usable also by clinicians. 

Author Response

Please find our responses in the attached word file.

Thanks!

Round 2

Reviewer 1 Report

The authors properly addressed my previous concerns

Reviewer 2 Report

I have no additional comments